# Identification of Movements and Postures Using Wearable Sensors for Implementation in a Bi-Hormonal Artificial Pancreas System

**DOI:** 10.3390/s21175954

**Published:** 2021-09-05

**Authors:** Ben Sawaryn, Michel Klaassen, Bert-Jan van Beijnum, Hans Zwart, Peter H. Veltink

**Affiliations:** 1Department of Biomedical Signals and Systems, Faculty of Electrical Engineering, Mathematics and Computer Science, University of Twente, P.O. Box 217, 7500 AE Enschede, The Netherlands; b.j.f.vanbeijnum@utwente.nl (B.-J.v.B.); p.h.veltink@utwente.nl (P.H.V.); 2Department of Research and Development, Inreda Diabetic B.V., 7472 DD Goor, The Netherlands; michel.klaassen@inredadiabetic.nl; 3Department of Applied Mathematics, Faculty of Electrical Engineering, Mathematics and Computer Science, University of Twente, P.O. Box 217, 7500 AE Enschede, The Netherlands; h.j.zwart@utwente.nl

**Keywords:** artificial pancreas, classification algorithms, inertial sensing, posture identification, movement identification, type 1 diabetes mellitus, wearable sensors

## Abstract

Background: Closed loop bi-hormonal artificial pancreas systems, such as the artificial pancreas (AP™) developed by Inreda Diabetic B.V., control blood glucose levels of type 1 diabetes mellitus patients via closed loop regulation. As the AP™ currently does not classify postures and movements to estimate metabolic energy consumption to correct hormone administration levels, considerable improvements to the system can be made. Therefore, this research aimed to investigate the possibility to use the current system to identify several postures and movements. Methods: seven healthy participants took part in an experiment where sequences of postures and movements were performed to train and assess a computationally sparing algorithm. Results: Using accelerometers, one on the hip and two on the abdomen, user-specific models achieved classification accuracies of 86.5% using only the hip sensor and 87.3% when including the abdomen sensors. With additional accelerometers on the sternum and upper leg for identification, 90.0% of the classified postures and movements were correct. Conclusions: The current hardware configuration of the AP™ poses no limitation to the identification of postures and movements. If future research shows that identification can still be done accurately in a daily life setting, this algorithm may be an improvement for the AP™ to sense physical activity.

## 1. Introduction

Closed loop bi-hormonal artificial pancreas systems, such as the artificial pancreas (AP™) developed by Inreda Diabetic B.V. (Goor, The Netherlands), fully regulate blood glucose levels of type 1 diabetes mellitus (T1DM) patients by administration of insulin and glucagon using a closed loop regulator as shown in Figure 1 [1]. The hardware consists of a mainframe, generally worn on the hip, two transmitters that measure blood glucose levels, worn in the abdomen region, and two cannulas through which insulin and glucagon are administered [2]. Furthermore, the mainframe and transmitters contain tri-axial accelerometers that sense physical activity [2] using a sampling frequency of 20 Hz as a measure of the metabolic consumption rate to adjust the hormone administration algorithm.

Because performing physical activity influences blood glucose levels based on the intensity of the activity [3], the AP™ control system allows for changing insulin administration levels at different activity magnitudes [2]. However, the current physical activity sensing method can be improved as the current implemented method of data processing entails heavily down-sampled 1D data, which will lead to a loss of information. Furthermore, other factors that are of influence on blood glucose dynamics, such as sleeping [4], are not taken into account in the current system as it does not know when a specific patient actually is sleeping. Finally, using the AP™ system in daily life means that the mainframe and transmitters are not placed consistently on the same location, where the mainframe is moved throughout the day due to natural use and the skin underneath the transmitters needs to recover from the adhesives. All these factors suggest that more detailed identification of physical activity using this particular setup is not trivial. As the AP™ currently does not classify postures and movements to estimate metabolic energy consumption to correct hormone administration levels, considerable improvements to the system can be made. To change the hormone administration algorithm based on the metabolic consumption rate more accurately, as a first step, postures and movements need to be identified as these individual identified postures and movements are related to specific metabolic consumption rates [5].

The concept of posture and movement identification using accelerometry has been researched before in a varying range of manners [6,7,8,9,10]. This knowledge has been applied in the diabetic field, for example by Dasanayake et al. [11]; by using an activity tracker worn on the wrist, they aimed to identify the onset and end of activities of a T1DM population before changes in blood glucose levels were induced. Jacobs et al. [12] utilized an accelerometer and a heart rate sensor on the sternum to detect activity by estimation of the energy expenditure of T1DM patients and proposed a method to adjust hormone administration levels based on the identified activities.

However, these studies make use of sensor placements that do not coincide when using the AP™ system. It is unknown whether solely using a sensor combination placed on the hip and abdomen region still enables algorithms to identify postures and movements. Furthermore, it has not been analyzed before how the performance of an identification algorithm may be affected when accelerometers are placed on different locations of a specific body segment after model development. Finally, even though activity recognition has been applied in the diabetic field [11,12], postures were not taken into account, which are important to consider [4].

Therefore, this research aims to investigate the possibility to use the current setup of the AP™ to identify several postures and movements using accelerometry as a first step in improving the hormone administration algorithm of the AP™ with respect to physical activity. During this research, we also investigated what the influence of varying locations of the sensors in the abdomen and hip regions was and the benefit of including additional sensors in the system was analyzed. The developed models for this identification algorithm should be as computationally sparing as possible due to the original system operating on batteries. Finally, for usability, the model should be as generalizable as possible, meaning a one-size-fits-all model structure was preferred if it was shown to perform better compared with tailor-made models.

## 2. Materials and Methods

### 2.1. Participants

Seven healthy human participants were recruited from the University of Twente (Enschede, The Netherlands) and Inreda Diabetic B.V. (Goor, The Netherlands). Participants who were eligible for participation were required to meet all the following criteria: (1) were over 18 years of age; (2) had no physical impairments; (3) had no cognitive disabilities; (4) and did not take medication, drugs, or excessive alcohol that may influence the performance of the experiment. This research was approved by the ethical committee of the Electrical Engineering, Mathematics and Computer Science faculty of the University of Twente (RP 2020-135) and all participants provided written informed consent prior to the start of this study.

### 2.2. Instrumentation

As can be seen in Figure 2, participants wore seven inertial measurement units (IMUs) (NGIMU, x-io Technologies, Bristol, UK), which measured acceleration using a sampling frequency of 20 Hz. One was placed on the upper left leg, one on the sternum, one on the left side of the pelvis, three on the abdomen (left and right of the umbilical region, near the edges of the lumbar areas, and one in the middle of the left lumbar region), and one on the left side on the body (the upper left lumbar region, near the left hypochondrium). The locations of the five IMUs in the abdomen region correspond to realistic varying positions of the sensors as placed when using the AP™ in daily life. The two other IMUs were expected to provide additional information regarding postures as the sternum and upper leg vary more in orientation when adopting several postures [8].

### 2.3. Experimental Setup

Figure 3 illustrates the floorplan of the experiment. Six postures and movements were taken into consideration to be identified, as many (instrumental) activities of daily living consist of (combinations of) these postures and movements: lying, sitting, standing, walking, running, and cycling [13,14]. The experiment consisted of three phases: performing isolated postures and movements, performing sequences of these postures and movements, and performing functional tasks. The load of the home trainer was set in such a way as to be comfortable for each individual participant.

#### 2.3.1. Isolated Postures and Movements

The summary of this phase is shown in Table 1. Participants were instructed to adopt specific postures and movements where each posture consisted of several variants. If a trial concerned postures, each posture variant had to be held for 20 s, whereas movements had to be performed twice for 30 s. All variants of a single posture were held once during each measurement trial and every possible permutation was measured. As the standing posture only concerns two variants and thus each variant is performed less often compared to lying and sitting, the standing trials were performed twice.

#### 2.3.2. Posture and Movement Sequences

During this phase, participants were instructed to perform sequences of postures and movements in a specific order. If a posture was held, the participant was free to take the position that he/she preferred, as long as it was one of the variants of that specific posture from the first experimental phase. The participants were free to perform the walking, running, and cycling tasks at the speed they preferred. Table 2 describes the performed sequences.

#### 2.3.3. Functional Tasks

The last phase of the experiment consisted of performing several functional tasks as they could occur during activities of daily living [13,14]. Participants were instructed to perform specific tasks, where the starting- and endpoints were clear, but they were free to perform it in a manner of their own choice. Table 3 describes the performed functional tasks.

### 2.4. Sensor-to-Segment Calibration

A sensor-to-segment calibration was performed prior to data analysis to represent the sensor data in a coordinate system that relates to the segment it is attached to, regardless of orientation [15]. Performing this calibration is beneficial as the effect of varying the orientation of sensors is minimized. The coordinate system of each segment was defined with respect to the anatomical position. Therefore, the *x*-axis was defined parallel to the anteroposterior axis pointing anterior, the *y*-axis parallel to the mediolateral axis pointing

Laterally left, and the *z*-axis parallel to the longitudinal axis pointing in the cranial direction. The direction of the *x*-axis was determined by measuring the gravitational acceleration when participants were lying down and the direction of the *z*-axis was determined by the measured gravitational acceleration during a quiet stance. Next, using the cross product of the former two axes, the *y*-axis direction was determined. Finally, the direction of the *x*-axis was recalculated using the cross product of the y- and *z*-axis to make the coordinate system orthonormal. These newly defined axes function as the rotation matrix and the raw acceleration data were transformed by matrix multiplication to be represented with respect to these previously defined axes.

### 2.5. Identification Algorithm

As the AP™ is a fully transportable system, it operates on batteries that last approximately 3 days and its transmitters last 3 weeks. Therefore, the software is carefully crafted to prevent it from consuming too much power. This means the algorithm should be as simple and computationally sparing as possible. Figure 4 is a schematic representation of the proposed identification algorithm, which can be implemented for any number and combination of sensors. First, data of each sensor are buffered for 10 s, to be used for feature extraction and data merging. Next, the merged features of all sensors are used for a distinction to determine whether an individual is assuming a posture or is moving within this 10-s window. Finally, these and other features are used to classify a specific posture or movement.

Two different types of models were fitted: user-specific models and generalized models. The former implies that data of each individual user were utilized to train and test the model such that the model performance for that specific user is maximized. The latter means that models were trained and tested using a leave-one-out training paradigm such that only one model is required for the entire population. The data acquired from the performed postures and movements as described in Section 2.3.1 were used to train the algorithm and the data acquired from the experiments described in Section 2.3.2 and Section 2.3.3 were used for model assessment. By considering a separate training and test set, overfitting for both types of models can be prevented.

#### 2.5.1. Posture Versus Movement Determination

For each individual sensor, first the 10-s buffered accelerometer data were used to calculate the standard deviation of the norm (StdNorm) every 1 s. As an initial estimate, each calculated StdNorm will be compared to a threshold, defined as the minimum calculated StdNorm value during the performed movements of the first phase. To finally determine whether a movement was measured in this timeframe of 10 s by a specific sensor, at least eight calculated StdNorm values had to be above the threshold, corresponding to at least 7.5 s of movement. If not, it was assumed a posture was measured during the last 10 s.

To distinguish postures from movements, the result of all sensors was combined and voting took place. For the algorithm to detect a movement, at least half of the sensors had to measure activity.

#### 2.5.2. Classification of a Specific Posture or Movement

Decision trees were trained in MATLAB using the *fitctree* function in its default settings to determine the optimal thresholds to classify between specific postures or movements. To identify the specific posture, the mean of the acceleration of the *x*- and *z*-axis was calculated every second to determine the bodysegment orientation. To classify between movements, the StdNorm was reused as a measure of activity level.

Within the timeframe of 10 s, using the previously mentioned features, a total of 10 classifications were performed, after which the final posture or movement was classified by voting. If there was a tie between classifications, e.g., five times walking and five times running, the classification of the previous timeframe was used as the decisive vote.

### 2.6. Resampling of the Ground Truth

The data were labeled in real time (20 Hz) using a stopwatch as ground truth. As the model is designed to identify postures and movements at 0.1 Hz, the ground truth was altered to match these time intervals. Therefore, within the 10-s intervals, the most performed posture or movement was used as the sole ground truth for that interval.

### 2.7. Assessment of Model Performance

Several situations were analyzed using the posture and movement identification algorithm to investigate whether the sensors present in the AP™ are sufficient or additional sensory information is required to identify postures and movements using the proposed algorithm. With seven sensors, several combinations were made to assess the effect of varying the sensor placement.

As described in Figure 2, the sensor locations were coded with the following numbers: 1 = Sternum, 2 = Upper left leg, 3 = Left hip, 4 = Right umbilical, 5 = Left umbilical, 6 = Left lumbar, and 7 = Upper left lumbar. Sensors 1 and 2 are locations that are normally not present when measuring with the AP™. Sensor location 3 is a realistic position of the mainframe, i.e., the AP™ sensor, sensor locations 6 and 7 are realistic positions of the wireless glucose sensors containing the accelerometers, i.e., the transmitters (the sensors coded with the circle and triangle in Figure 1), and sensor locations 4 and 5 are realistic positions of either the AP™ sensor or transmitters.

Finally, as stated in Section 2.5, to assess robustness, these models not only were developed to be user specific, but also adopted a generalized form using a leave-one-out training paradigm. Statistical analyses were performed in IBM SPSS (v27) using an ANOVA with Bonferroni post hoc tests to investigate whether there are significant differences in model performance between user-specific versus generalized models and between using different sensor combinations as explained in the following sections.

#### 2.7.1. Assessment of Normal Usage of the AP™ System

The first analysis investigates the effect of including the information of the transmitters when using the AP™ normally, as illustrated in Figure 1. This means sensor 3 was used as the AP™ sensor and sensors 4 and 5 were used as transmitters (i.e., the accelerometers within the sensors depicted by a circle and triangle in Figure 1) during this analysis. Models were trained and tested on the situations where only the AP™ is used (3), the AP™ is used with one transmitter (3, 4 and 3, 5), and the AP™ is used with two transmitters (3, 4, 5).

#### 2.7.2. Inclusion of Additional Sensors in the AP™ System

This analysis provides information of the benefit of including more sensors than currently used in the AP™ system, i.e., the sensors on the sternum (1) and/or left leg (2). In terms of accuracy, the best-performing sensor (combination) of the previous analysis phase (Section 2.7.1) was used as a basis. This means models were trained and tested using just data from the sternum and left leg sensors, using the sternum or left leg sensor with the best sensor (combination) of the previous analysis, and using the sternum and left leg sensors with the best sensor (combination) of the previous analysis.

#### 2.7.3. Varying Sensor Placement after Model Training

Using the AP™ system in daily life means that the AP™ and transmitters are not placed consistently on the same location, where the mainframe is moved throughout the day and transmitters are moved once every four to six days. To investigate the effect of varying the sensor placement of the AP™ and/or transmitters, models that were previously trained from the first analysis phase (Section 2.7.1) were used to test a different dataset. To analyze the effect of varying the AP™ sensor placement, the trained model of data of the AP™ sensor (3) was tested on the sensor data of locations 4 and 5. To analyze the effect of varying the transmitter placement, the trained model of the AP™ sensor and both transmitters (4 and 5) was tested on data using the same AP™ sensor, but with varying locations of the transmitters (4, 5, 6 and/or 7). The combined situation was also assessed, meaning the trained model of the AP™ sensor and both transmitters was tested on a dataset when all three sensor locations were varied.

## 3. Results

### 3.1. Dataset

All participants fully completed all experiments; however, due to sensor battery problems and errors, sensor 6 did not measure during the functional tasks of subject 1 and the first sequence 1 was not measured of subject 2.

### 3.2. Model Performance Using Normal AP™ Sensor Placement

Table 4 summarizes the classification results of all seven subjects when using models developed with data of the AP™ sensor with or without transmitter sensors. Here, accuracy is defined as the total correct classified postures and movements as a percentage of the total performed postures and movements of all participants combined. The highest mean classification accuracy of 87.3% was achieved with user-specific models using the AP™ with both transmitters (i.e., combination 3, 4, 5), which is an increase of 0.8% when using just the AP™ (represented by sensor 3). However, using different sensor combinations for model development in these cases does not significantly change classification accuracies. Furthermore, developing generalized models significantly decreases accuracy compared with user-specific models.

Figure 5 shows the confusion matrices of all subjects combined after assessing each user-specific model’s performance using several sensor combinations. Overall, positive predictive values of at least 94.7% for lying, walking, and running were achieved for sensor combinations 3 and 3, 4, 5. Noticeable classification errors for postures either using sensor combination 3 or 3, 4, 5 were observed for sitting classified as standing (93 and 61 occurrences, respectively) and standing classified as sitting (30 and 37 occurrences, respectively). Analyzing the matrices regarding movements, misclassification for cycling is observable for both sensor combinations with positive predictive values of 71.4% and 69.0%, respectively.

### 3.3. Model Performance Using the AP™ and Additional Sternum and/or Upper Leg Sensors

As the previous analysis showed no significant differences in classification performance between sensor combinations, this analysis was performed using solely the AP™ sensor (3) with additional sternum (1) and/or upper leg sensors (2).

Results are presented in Table 5. Combining the sensor on the sternum with the AP™ sensor decreases accuracy significantly compared with using the AP™ sensor by itself. Furthermore, including both sternum and upper leg sensors increases the classification accuracy up to a maximum performance of user-specific models at 90.0%, versus 86.5% when using just the AP™ sensor data. This result, however, was not shown to be significant.

The confusion matrices in Figure 6 show positive predictive values of at least 90.6% for each posture/movement, except cycling, for sensor combinations 1, 2 and 1, 2, 3. Classification errors for lying and sitting were observed less when using these combinations compared with the other two confusion matrices. However, misclassification for cycling is still observable for both sensor combinations 1, 2 and 1, 2, 3 with positive predictive values of 57.3% and 74.2%, respectively.

### 3.4. Model Performance Using Varying AP™ Sensor Placement after Model Training

Table 6 shows the influence of varying the AP™ sensor placement after the model has been trained on data of the standard AP™ sensor placement (3). In general, changing the sensor location after model training decreases classification accuracies significantly.

Varying the transmitter sensor placement (4, 5, 6, and/or 7) whilst using the same location for the AP™ sensor (3) does not significantly in- or decrease classification accuracies, as can be seen in Table 7.

As shown in Table 8, the combined effect of both alterations was also observed, where changing the AP™ sensor location decreases the classification accuracy and varying the transmitter sensor placement does not show an in- or a decrease in accuracy. However, due to the added information of the transmitter sensors, the decrease in model performance due to varying the AP™ sensor placement was not found to be significant anymore.

## 4. Discussion

This research aimed to investigate the possibilities to identify postures and movements of human subjects using the current hardware design of the artificial pancreas (AP™) system. Using the standard AP™ and transmitter sensor placement whist sampling at 20 Hz, user-specific models achieved classification accuracies of 86.5% without transmitters and 87.3% with transmitters. Comparing these results with a sensor combination including additional sensors for identification, a nonsignificant result of 90.0% accuracy was achieved. Therefore, from these results it can be said that the current hardware configuration either with or without using transmitters poses no limitation to performing the proposed classification task and that including extra sensors in the system may be beneficial for classification accuracy in terms of improved posture recognition.

Even though this research cannot be exactly compared, generally speaking this identification algorithm performs in a manner comparable to or better than other studies. However, the reported studies were limited in the sense that not all mentioned postures and movements (i.e., lying, sitting, standing, walking, running, and cycling) were classified in cases that were most similar to this research. Overall, the research by Curone et al. [9] reported an average accuracy of 96.2% using a sensor on the sternum and a simplified classification paradigm by assigning several postures and activities to one class. For example, standing upright and sitting down were assigned to the class “upright standing”. This means their model performance is expected to be diminished without this simplification, as was the case during our research. Furthermore, Fortune et al. [10] reported being unable to distinguish sitting from lying using an accelerometer on the waist, which during our research was possible with a positive predictive value of 74% using a sensor on the hip. Their research reports positive predictive values for lying, standing, walking, and running of 97%, 71%, 76%, and 100%, which our algorithm outperforms with positive predictive values of 99.8%, 84.8%, 94.7%, and 98.2%, respectively, when using a sensor on the hip.

Specific placement of the AP™ sensor after training the models is critical. Training and testing user-specific models based on data of sensor location 3 resulted in a classification accuracy of 86.5%, whereas testing on data of sensor locations 4 or 5 resulted in significantly lower classification accuracies of 78.1% and 75.6%, respectively. On the other hand, varying transmitter locations have shown to be of no significance, where the highest and lowest classification accuracies of 87.3% and 85.3%, respectively, were observed. This result suggests that changing transmitter locations after training does not significantly alter the model’s performance. Furthermore, a significant decrease in classification accuracy due to varying the AP™ sensor placement was no longer observed when data of the transmitters were included, as can be seen in Table 8. Thus, data from the transmitters, placed on the abdomen, will provide a stable factor for the identification algorithm when combined with data from the AP™ sensor.

For easier implementation, it is preferred to have a generalized model that is applicable to a broader user spectrum. Comparing results between user-specific and generalized models indicates that performance significantly decreases when adopting a generalized form in all cases. It should therefore be carefully considered whether the loss in classification accuracy is acceptable when the algorithm is implemented in the generalized form.

The experiment was performed in a controlled lab environment that does not naturally translate to recognition of postures and movements when performing actual daily life activities for a prolonged period. During the experiment, the least active movement measured was cycling, as it was performed on a statically placed home trainer. This affects the minimum threshold to distinguish postures from movements, which is expected to be lower compared with performing daily life activities. This means the reported classification accuracies might be higher when accounting for the minimum threshold during movements in daily life, which is expected to be higher compared with performing movements in an experimental setting. Furthermore, the experiment consisted of short posture and movement sequences, where the subject had to change from one to another in quick succession. These transitions pose a challenge for the algorithm as during a 10-s window, several postures and/or movements are performed and thus the chance of misclassification is higher compared with performing one posture or movement during this 10-s window. During daily life, these transitions are expected to occur less frequently than during the experiment, meaning the algorithm is expected to perform better during a daily life implementation.

Prior to the experiment, subjects were subjected to a calibration phase, which has to be performed to represent the sensor data in a coordinate system that relates to the segment it is attached to, regardless of orientation [15]. Even though this is beneficial for model performance, an explicit calibration is unfeasible when the algorithm is implemented for daily life as the AP™ sensor may vary in location throughout the day and transmitter sensor locations after a few days. Performing a calibration after any sensor adjustment may be tiresome for the end-user. Therefore, it should be investigated whether an autocalibration algorithm without initial information regarding sensor placement is a solution to this issue. By performing a continuous implicit sensor-to-segment calibration, the end-user will not be burdened with this task each time the position of a sensor is changed.

The final factor that should be considered when implementing this algorithm in the AP™ system is the fact that standing and sitting are difficult to distinguish without using additional sensors 1 and 2, as seen in Figure 5 versus Figure 6. Based on the interests of the end-user and manufacturer, these two postures could be regarded as one as they are expected to have no different influence on blood glucose levels. Lying is a key posture, as it is the general posture a human adopts when sleeping, which may influence blood glucose levels via the dawn phenomenon [4] and should thus be regarded separately. This simplification of classification should also improve classification accuracies.

To validate these expected performance changes of the algorithm, longitudinal human activity data using a wearable sensor setup should be gathered, which in turn are to be used for the analysis of posture and movement recognition in a daily life setting. In addition to this activity classification, an estimation of the metabolic consumption rate is to be performed as different activity magnitudes are related to different changes in blood glucose levels [3]. In the case of performing movements, the metabolic costs may relate to the standard deviation and frequency content of the accelerometer signals [5,16]. Therefore, additional (literature) research has to be performed to investigate how postures and movements relate to metabolic energy consumption and thus blood glucose levels, such that ultimately the blood glucose regulation in the AP™ can be changed according to this posture and movement identification algorithm.

## 5. Conclusions

This study showed that the current hardware configuration of the artificial pancreas (AP™), developed by Inreda Diabetic B.V., poses no limitations to identifying specific postures and movements of human subjects. Using accelerometers placed on the hip and abdomen that sample at a sample frequency of 20 Hz, user-specific and computationally sparing models correctly classified postures and movements with an accuracy of 87.3%. When the model adopted a generalized form, classification accuracies significantly decreased and thus the algorithm may be more suitable for user-specific training. Furthermore, if used by itself, the exact placement of the accelerometer on the hip is of significant influence for classification performance and including additional sensors on the sternum and/or upper leg may significantly improve classification accuracies. If future research shows that the proposed algorithm is still able to accurately identify postures and movements in a daily life setting, a first step in improving the hormone administration algorithm of the AP™ to account for changes in physical activity will have been made.

## Figures and Tables

**Figure 1 sensors-21-05954-f001:**
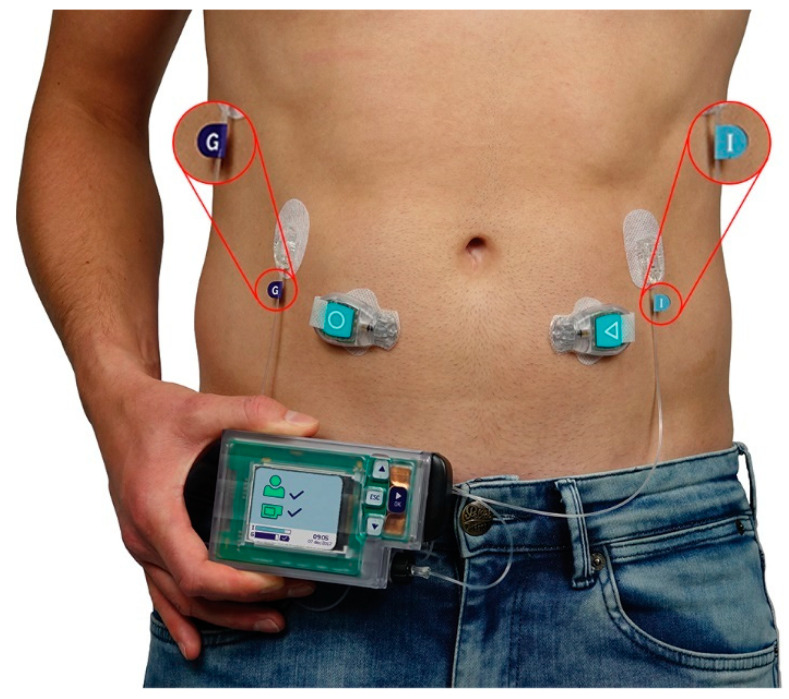
Representation of the AP™ as it generally is worn by T1DM patients. Here, the mainframe, as shown in the subject’s right hand, at least consists of the main processor, regulatory hormones, and an accelerometer. Furthermore, the transmitters, depicted by the devices with a circle and triangle, contain the blood glucose sensors and accelerometers. Finally, the cannulas (purple G and blue I) serve as an injection spot for the regulatory hormones [1].

**Figure 2 sensors-21-05954-f002:**
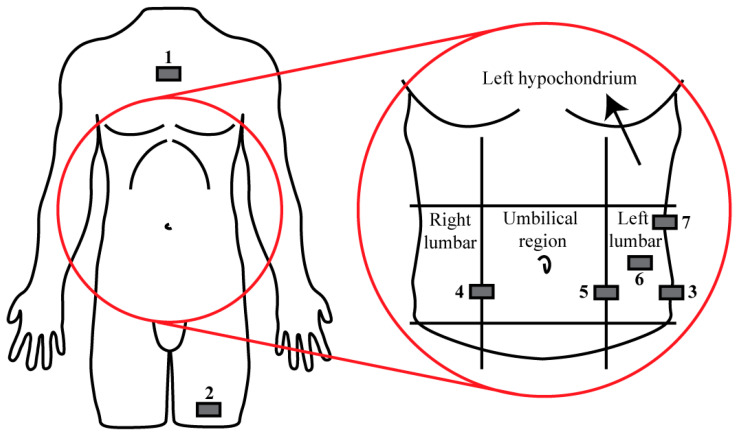
Schematic representation of the locations of the IMU placements (grey rectangles). Throughout the experiment, the sensor locations were coded with the following numbers: 1 = Sternum, 2 = Upper left leg, 3 = Left hip, 4 = Right umbilical, 5 = Left umbilical, 6 = Left lumbar, and 7 = Upper left lumbar.

**Figure 3 sensors-21-05954-f003:**
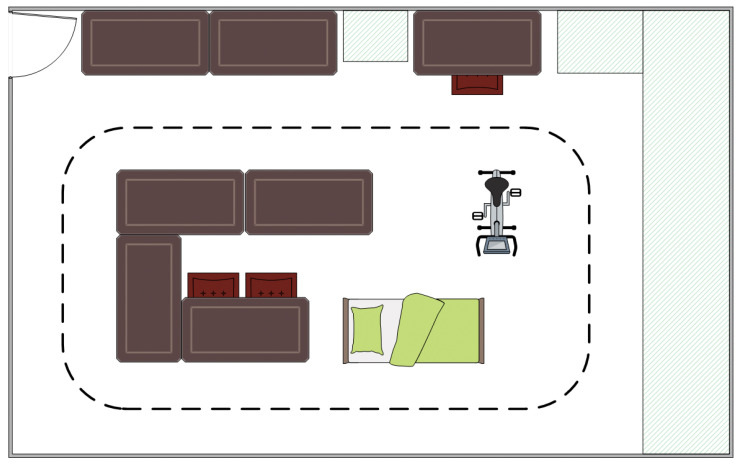
Sketch of the experimental floorplan (not to scale). The dotted line represents the walking and running course. The illustrated bed was a treatment table, the chairs in the middle allowed participants to sit down, and the home trainer allowed for cycling activities.

**Figure 4 sensors-21-05954-f004:**
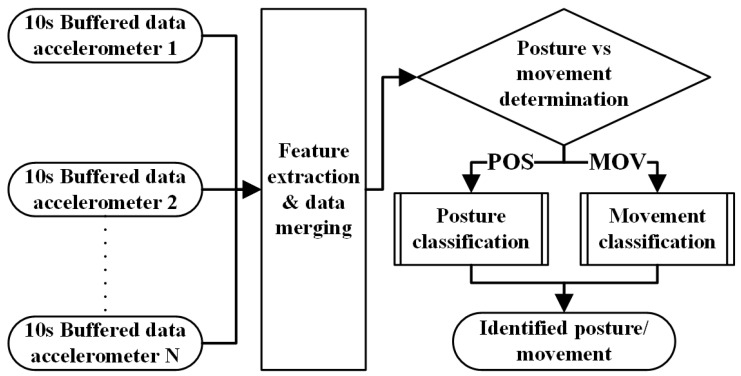
Schematic representation of the identification algorithm, which is applicable for any number and combination of sensors. To distinguish postures from movements in general, the standard deviation of the norm was utilized. This same feature was used to classify a specific movement. Finally, the mean value of the *x*- and *z*-axis of each sensor was used as a feature to classify a specific posture.

**Figure 5 sensors-21-05954-f005:**
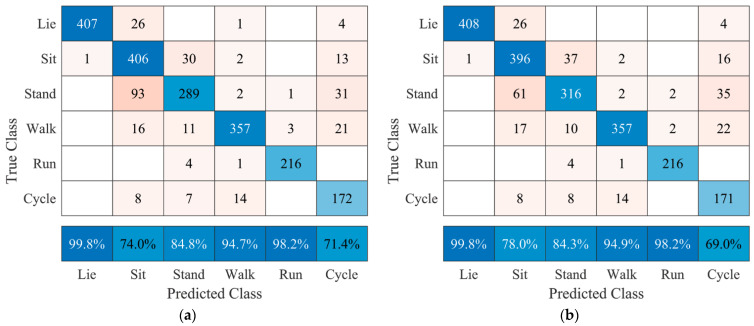
Confusion matrices after testing user-specific models using several sensor combinations of all seven subjects. (**a**) only sensor 3 (accuracy 86.5%); (**b**) sensors 3, 4, and 5 (accuracy 87.3%). The bottom row represents the positive predictive value for each posture or movement.

**Figure 6 sensors-21-05954-f006:**
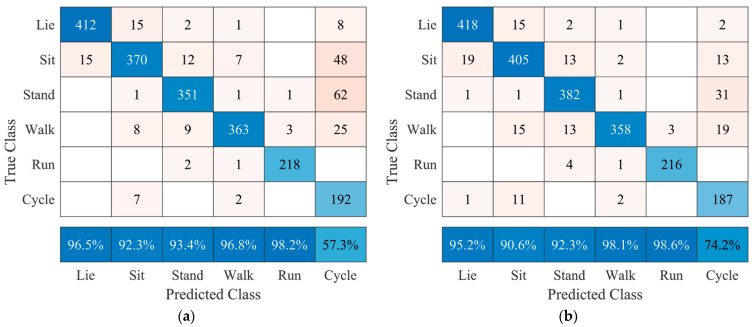
Confusion matrices after testing user-specific models using several sensor combinations of all seven subjects. (**a**) sensors 1 and 2 (accuracy 89.2%); (**b**) sensors 1, 2, and 3 (accuracy 90.0%). The bottom row represents the positive predictive values for each posture or movement.

**Table 1 sensors-21-05954-t001:** Summary of the performed trials of the first phase. The postures consisted of several variants and the movements were performed at the preferred speed of each participant.

Posture/Movement	Variants	Number of Trials
Lying	Lying on their back, right side, or left side.	6
Sitting	Sitting on the edge of the chair, using the backrest normally, or slouched.	6
Standing	Standing upright or in a bending forward position	4
Walking	None	2
Running	None	2
Cycling	None	2

**Table 2 sensors-21-05954-t002:** Description of the performed sequences. Abbreviations: L = Lie, Si = Sit, St = Stand, W = Walk, R = Run, C = Cycle. The first sequence was performed twice, all other sequences once.

Sequence	Description
1	L (60 s)–Si (30 s)–St (30 s)–W (2 laps)–Si (30 s)–W (1 lap)–R (3 laps)–W (1 lap)–C (20 rot.)–St (50 s)–W (5 s)–Si (10 s)–L (60 s).
2	W (1 lap)–R (3 laps)–W (1 lap)–C (20 rot.)–Si (30 s)–St (30 s)–W (2 laps)–Si (30 s)–L (60 s)–St (50 s)–W (5 s)–Si (10 s)–W (1 lap)–R (3 laps)–W (1 lap)–C (20 rot.).
3	L (60 s)–St (50 s)–W (5 s)–Si (10 s)–W (1 lap)–R (3 laps)–W (1 lap)–C (20 rot.)–Si (30 s)–St (30 s)–W (2 laps)–Si (30 s)–W (1 lap)–R (3 laps)–W (1 lap)–C (20 rot.).
4	L (40 s)–Si (20 s)–L (10 s)–St (15 s)–W (4 laps)–Si (5 s)–St (5 s)–R (8 laps)–St (20 s)–W (2 laps)–Si (5 s)–L (10 s)–Si (30 s)–L (10 s).
5	L (20 s)–Si (20 s)–L (20 s)–Si (20 s)–St (10 s)–R (7 laps)–St (40 s)–Si (30 s)–St (30 s)–L (40 s)–Si (20 s)–St (20 s)–W (1 lap)–Si (20 s)–L (60 s).

**Table 3 sensors-21-05954-t003:** Description of the functional tasks that were all performed once.

Functional Task	Description
Go to work	Participants started sitting anywhere and then either walked or ran for 10 laps or cycled for 100 rotations. The task ended when the participant sat on a chair at the table.
Tidying	Participants had to put several items, which were spread on a table, in a crate. After completion, they had to go to a different table and the task ended after unloading all items from the crate on the table.
Interval training	A sporting exercise where participants had to walk (5 laps), run (10 laps), and cycle (100 rotations) at a speed and in an order they preferred. After each exercise, there was a resting period of 30 s.
Nightly toilet visit	Participants started lying down for a duration between 15 and 30 s, after which they had to walk to a seat and sit down for a duration between 15 and 30 s. The task was finished when they lay down again.
Obstacle course	Participants had to walk back and forth five times on a straight course, which consisted of an obstacle the participants had to step over and a narrow space (two tables close to each other) participants had to walk through.

**Table 4 sensors-21-05954-t004:** Classification results for different sensor combinations when using the AP™ normally. Shown are mean accuracies (%) and the corresponding standard deviation of all seven subjects when developing user-specific or generalized models.

Sensors	User Specific *	Generalized *
3	86.5 ± 5.2	76.9 ± 7.9
34	87.1 ± 3.4	74.6 ± 7.9
35	87.2 ± 3.1	75.2 ± 8.1
345	87.3 ± 3.0	75.3 ± 7.9

Conditions that share the same symbol (*) differ significantly in terms of model performance; *: *p* < 0.001.

**Table 5 sensors-21-05954-t005:** Classification results for different sensor combinations when using the AP™ and additional sternum and/or upper leg sensors. Shown are mean accuracies (%) and the corresponding standard deviation of all seven subjects when developing user-specific or generalized models.

Sensors	User Specific *	Generalized *
3 ^¤^	86.5 ± 5.2	76.9 ± 7.9
12 ^ǂ^	89.2 ± 4.8	70.2 ± 9.6
13 ^¤, §, ~^	81.6 ± 8.2	64.0 ± 7.5
23 ^§^	82.1 ± 5.7	84.2 ± 4.9
123 ^ǂ, ~^	90.0 ± 5.0	87.4 ± 5.0

Conditions that share the same symbol (*, ¤, ǂ, §, ~) differ significantly in terms of model performance; ^~^: *p* < 0.001; ^§^: *p* < 0.005; ^¤^, ^ǂ^: *p* < 0.02; *: *p* < 0.05.

**Table 6 sensors-21-05954-t006:** Classification results for different sensor combinations when assessing model performance using varying sensor placements after training the model based on the data of sensor location 3. Shown are mean accuracies (%) and the corresponding standard deviation of all seven subjects when developing user-specific or generalized models.

Sensors	User Specific *	Generalized *
3 ^¤,^ ^ǂ^	86.5 ± 5.2	76.9 ± 7.9
4 ^¤^	78.1 ± 6.6	68.7 ± 7.9
5 ^ǂ^	75.6 ± 7.8	72.8 ± 5.6

Conditions that share the same symbol (*, ¤, ǂ) differ significantly in terms of model performance; *: *p* < 0.005; ^¤^, ^ǂ^: *p* < 0.05.

**Table 7 sensors-21-05954-t007:** Classification results for different sensor combinations when assessing model performance using varying transmitter sensor placements after training the model based on the data of sensors 3, 4, and 5. Shown are mean accuracies (%) and the corresponding standard deviation of all seven subjects when developing user-specific or generalized models.

Sensors	User Specific *	Generalized *
345	87.3 ± 3.0	75.3 ± 7.9
346	85.3 ± 7.7	77.5 ± 9.3
347	85.3 ± 7.6	77.4 ± 9.3
356	85.3 ± 7.5	79.6 ± 8.1
357	85.4 ± 7.5	79.5 ± 7.8
367	85.3 ± 7.4	78.1 ± 7.1

Conditions that share the same symbol (*) differ significantly in terms of model performance; *: *p* < 0.001.

**Table 8 sensors-21-05954-t008:** Classification results for different sensor combinations when assessing model performance using varying AP™ and transmitter sensor placements after training the model based on the data of sensors 3, 4, and 5. Shown are mean accuracies (%) and the corresponding standard deviation of all seven subjects when developing user-specific or generalized models.

Sensors	User Specific *	Generalized *
345	87.3 ± 3.0	75.3 ± 7.9
456	77.1 ± 8.7	74.2 ± 9.2
457	77.0 ± 8.5	72.5 ± 8.2
467	77.2 ± 8.5	70.9 ± 5.5
567	74.9 ± 8.9	73.2 ± 3.9

Conditions that share the same symbol (*) differ significantly in terms of model performance; *: *p* < 0.005.

## Data Availability

Data are available from the authors upon request.

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
