# Peer review of "Identification of Movements and Postures Using Wearable Sensors for Implementation in a Bi-Hormonal Artificial Pancreas System"

_sensors, 2021, doi:10.3390/s21175954_

Round 1

Reviewer 1 Report

In my opinion, this article does not show quantitatively the deficiencies of the AP system. Neither does it show quantitatively how the functioning of the AP improves thanks to the postural classification.

On the other hand, the originality of the posture classification method is not explained. The results are not compared to the state of the art.. In the introduction you should mention articles related to the posture classification, the originality of your work with respect to others, the main results of your work and a summary of the different sections of the article.

I don't understand why you conduct a study for different sampling frequencies.

You need to analyze whether your method generalizes correctly, perhaps using a cross validation procedure.

Reviewer 2 Report

  • This paper seems to support the hypothesis that some additional sensors to an existing platform [1] could improve the identification of human actions or postures in order to assist an artificial pancreas. Please feel free to correct me if this is not the case.
  • The abstract of the paper states in its conclusions that "this algorithm may be an improvement...". Section 2.5 poses an algorithm that I understand is a part of an existing system although with less sensors. At this moment doesn't seem clear to me if the paper is describing the whole system from an engineering stand point or shows enhancements to an existing platform.
  • Regarding the writing style, in  my humble opinion could be made more readable. Sencences like "(45) The APT. Shown is the mainframe in the subject his right hand, ..." need some rewriting effort.
  • Figure 1 displays a snapshot with labels too small to be seen even in printed format.
  • Figure 2 is very little informative and could be improved to include labels described in subsection 2.7 that are hard to discern only in the written description.
  • Section 2.5 referes to a feature extraction task and could be interesting to name precisely which are these features. ¿In the line (164) ( "the merged features of all sensors are used for a distinction..." ) could you be referring to pattern classification?
  • In Figure 4, "Determination posture vs movement" perhaps better could be "Posture vs. movement determination". Please use a proper symbol (diamond) to represent decission.
  • Subsection 2.7.1 uses the term "transmitter sensor" that seems different from the "hip sensor" and It has not sense to me if this sensor is located in the main unit of any other place. I understand that if you use the term "mainframe" for the pocket sized hardware there is no sensor included in it (foll. Merriam-Webster).
  • Some bibliography references appear to be a little bit outdated (only four references are in 2000s) and I think that it is worthwhile the effort broadening the search to improve your introduction section. The next references could serve as samples that could be useful among many other possible ones.
    • Ding, S. and Schumacher M. "Sensor Monitoring of Physical Activity to Improve Glucose Management in Diabetic Patients: A Review", Sensors 2016. 
    • Curone, Davide, et al. "A real-time and self-calibrating algorithm based on triaxial accelerometer signals for the detection of human posture and activity." IEEE transactions on information technology in biomedicine 14.4 (2010): 1098-1105.
    • Wong, Wai Yin, Man Sang Wong, and Kam Ho Lo. "Clinical applications of sensors for human posture and movement analysis: a review." Prosthetics and orthotics international 31.1 (2007): 62-75.
    • Llamas, César, et al. "Open source platform for collaborative construction of wearable sensor datasets for human motion analysis and an application for gait analysis." Journal of biomedical informatics 63 (2016): 249-258.

Finally, your conclusions are very timid and the main hypotheses and results are not very clear. In my opinion your paper is very interesting. Extending the introduction and detailing better your electronics, computer platform and algorithmics could enhance it easily. 

Round 2

Reviewer 1 Report

OK about some answers. I think your article has improved. However I have some additional comments:

  1. Regarding your study for different sample frequencies: In my opinion, it is not necessary. You just need to apply Shannon to find the minimum fs = 2fm (twice the maximum frequency of the signal). The selected fs allows you to work with the minimum sampling frequency without losing information.

  2. Concerning the model generalization; I think you need to improve some sentences:

The paragraph (L194) 'Section 2.3.1. were used to train the algorithm, the data acquired from the experiments described in Sections 2.3.2. and 2.3.3. were used for model assessment.'. After reading this sentence, I understood that you would like to analyze how well your model can be generalized by comparing a single training group and a single test group. This would be a good approach with a large amount of data.

However, using seven people, I think the amount of data is not huge. Consequently, one solution is to check the generalization using a cross-validation approach as you mention on line 256. I imagine that Figures 5 and 6 have been calculated using that cross-validation.

Later you wrote (L411): "For easier implementation it is preferred to have a generalized model which is applicable to a broader user spectrum. Comparing results between user specific and generalized models show that performance significantly decreases when adopting a generalized form. It should therefore be carefully considered whether the loss in classification accuracy is acceptable when the algorithm is implemented. " And in L471: "When the model adopts a generalized form, classification accuracies significantly decreased and thus the algorithm may be more suitable for user-specific implementations ".

I think, you should delete those paragraphs, why? Because the goal of a generalized model is to avoid overfitting and you should always implement a generalized model. A generalized model cannot be considered a possibility.  Also, if your model is generalized, I don't understand why you say ' be carefully considered whether the loss in classification accuracy is acceptable when the algorithm is implemented."
